# Evaluation of counting methods for monitoring populations of a cryptic alpine passerine, the rock wren (Passeriformes, Acanthisittidae, *Xenicus gilviventris*)

**Joanne M. Monks** [1]*, **Colin F. J. O'Donnell**[2], **Terry C. Greene** [2], **Kerry A. Weston**[2]

**1** Biodiversity Group, Department of Conservation, Dunedin, New Zealand, **2** Biodiversity Group, Department of Conservation, Christchurch, New Zealand

* jmonks@doc.govt.nz

**Data Availability Statement:** All relevant data are within the paper and its Supporting Information files.

## Abstract

Developing and validating methods to determine trends in populations of threatened species is essential for evaluating the effectiveness of conservation interventions. For cryptic species inhabiting remote environments, this can be particularly challenging. Rock wrens, *Xenicus gilviventris*, are small passerines endemic to the alpine zone of southern New Zealand. They are highly vulnerable to predation by introduced mammalian predators. Establishing a robust, cost-effective monitoring tool to evaluate population trends in rock wrens is a priority for conservation of both the species and, more broadly, as part of a suite of indicators for evaluating effectiveness of management in New Zealand's alpine ecosystems. We assessed the relative accuracy and precision of three population estimation techniques (mark-resight, distance sampling and simple counts on line transects) for two populations of rock wrens in the Southern Alps over six breeding seasons (2012–2018). The performance of these population estimators was compared to known rock wren population size derived from simultaneous territory mapping. Indices of abundance derived from counts on transects were correlated with territory mapping at both study areas, and performed better than either mark-resight methods or distance sampling. Simple counts on standardised line transects are a highly cost-effective method of monitoring birds because they do not require banding a population. As such, we recommend that line transect counts using the design outlined in this paper be adopted as a standard method for long-term monitoring of rock wren populations. Although species-specific testing is required to validate use of low-cost population indices, our results may have utility for the monitoring of other cryptic passerines in relatively open habitats.

## Introduction

Developing monitoring methods for threatened species that are difficult to detect (i.e. 'cryptic') can be problematic, particularly in challenging environments. However, information on

**Funding:** The New Zealand Department of Conservation (https://www.doc.govt.nz/) funded the work presented in this paper. The funder did not play any role in any aspect of the research or manuscript preparation, but all work was done by employees of the government department that funded the work.

**Competing interests:** All authors are employees of the government department that funded this work.

population trends is one of the metrics used in assessing species status (e.g. by the International Union for the Conservation of Nature) and is essential for evaluating the effectiveness of conservation interventions. Therefore, development and validation of an appropriate method to estimate abundance and derived population trends, that is achievable within budget constraints, is vital for all threatened species prioritised for management [1–4].

Absolute measures of population abundance and density are often extremely difficult and costly to obtain. There has been considerable recent debate over the assumptions and application of indices [5] as well as development of estimation methods that explicitly address concerns regarding variable detectability [6–8]. However, newer methods often involve restrictive assumptions, complex field designs and analyses and, therefore, high costs [9]. Cost-effective and robust methods that can be used to confidently detect population trends are essential for conservation managers [3,10].

Alpine ecosystems in New Zealand are under increasing pressure from the interacting effects of climate change, invasive browsers and predators [11]. Yet, information about alpine ecology is lacking. Despite encompassing 11% of the land mass of New Zealand [12], a dearth of biodiversity monitoring in the alpine zone means that no alpine taxa are included in national scale trend reporting [13]. The development of robust and logistically feasible monitoring methods for species above the timberline has been identified as an urgent requirement [11,12,14] so that a suite of alpine indicators can be used to measure trends, including response to management, in this nationally significant ecosystem.

Rock wrens (*Xenicus gilviventris*; also known as pīwauwau, mātuitui, and tuke) are small (14–20 g) passerines endemic to the alpine zone of New Zealand's South Island. They are poor fliers, and difficult to detect given their small size and the challenging high-altitude environment they occupy amongst alpine scrub, boulder fields and rocky bluffs [15]. Further, rock wrens have become increasingly rare, largely due to unsustainable predation by invasive mammalian predators [12,16]. Human-induced climate change is also predicted to place further pressure on rock wren populations occupying geographically disjunct mountain ranges [17,18]. The IUCN conservation status of Endangered reflects the ongoing threats to the species [18]. Recent genetic work also identified two distinct lineages within the population [17], the southern lineage is currently classified as Nationally Endangered, and the northern lineage as Nationally Critical [19]. Establishing a robust, cost-effective monitoring tool to evaluate long-term population trends in rock wrens is a priority for conservation of the species [18]. We aimed to develop a cost-effective method of monitoring rock wrens that would enable monitoring of population trends over time and ultimately assess the effectiveness of conservation interventions (landscape-scale predator control and translocations) with confidence.

## Methods

We reviewed potential monitoring techniques for rock wrens based on established bird counting techniques used for similar rare and cryptic taxa (Table 1). Based on this review, we decided to proceed with a comparison of three methods (mark-resight, distance sampling and simple counts on line transects) against territory mapping, the latter of which we considered the 'gold standard' [2,20]. Territory mapping enables estimation of population size in the breeding season when complete censuses are not possible [20,21]. Mark-resight and distance sampling involve incomplete counts that estimate absolute density and account for variable detection probabilities. Counts on line transects are indices that estimate relative abundance, but do not adjust for detection probabilities. We decided against trialling site occupancy as a method because detectability can vary with rock wren

**Table 1. Potential monitoring techniques considered for rock wrens based on techniques used for similar rare and cryptic taxa.**

| Method[a] | Main assumptions | Advantages for rock wrens | Disadvantages and biases for rock wrens | Cost | References |
|---|---|---|---|---|---|
| Territory mapping | Observer is good at finding and identifying birds. Records are plotted accurately. There is a reasonable chance of detecting every territory that overlaps with the defined sampling area Biases are standardised. | A thorough method for determining territories that would give a good indication of decline of rock wrens over time particularly in small populations. Do not need to mark individual rock wren (though this would give more accurate results). Method could be used to assess breeding success. | May only capture estimates of breeding population and not account for other individuals. Territories of rock wrens overlapping with the sampling area may be missed where birds primarily occupy bluff habitat and rarely use accessible habitat. Can be intensive and time consuming requires many repeat visits. Subject to variation in time, weather, observer abilities and so need to standardise these biases. | Medium to high. Repeat surveys can be time and resource consuming. | [20,22] |
| Mark-resight | Birds have same probability of been caught. Population is closed for the survey. Marks are permanent. | Highly precise result if assumptions are met. Analysis of data is straightforward with NOREMARK. Good for estimating other useful information such as survival, population trends and recruitment which is required for rock wrens. Closed population assumption should hold true for rock wrens during short sampling sessions. | May be difficult to obtain high level of re-sightings. Requires 40% of birds to be individually marked, which is time and resource consuming for rock wrens. Requires constant up-keep of colour banding if monitoring is to be long term. | High. Need to maintain a banded population. Banding birds is time and resource consuming. | [20,23] |
| Site occupancy | Sites remain occupied or unoccupied for duration of survey. Species are available for detection for duration of survey. | Only need to detect species once by sight, sound or other cues in each site. Cost effective and efficient method for covering large areas making it ideal for range and distribution. | Need to survey sites a number of times to improve accuracy of probability function. Cannot estimate population size from method. Rock wrens may not always be available for detection inside site (e.g. they are may travel up cliff faces). | Medium. Repeat surveys can be time and resource consuming. | [24,25] |
| Distance sampling (from point counts or line transects) | All birds on the line or point are detected. Birds do not move towards or away from the observer. Distance to observations are measured accurately. | Good method for robust, unbiased abundance estimation. Reduces the incomplete detectability resulting from simple counts. Estimates of population can be compared across time and habitat. No need to count all birds in area. | Violation of assumptions can lead to large errors. Minimum number of detections required. 60 for line and 80 for point transects. This may be difficult to obtain for rock wrens. Often only hear rock wren so may create inaccuracies in distance measurement. | Low to med. Requires no capturing or marking of birds. May need some observer training. | [6,26] |
| Simple counts (from point counts or line transects) | Sample points distribution random, systematic or stratified. Detection probabilities remain constant. Relationship between index and true abundance is linear. Must calibrate if using to estimate density. Birds not double counted. | Cost effective and efficient method for covering large areas of alpine habitat. Method may be more suited to alpine habitat than other more intensive methods as actual distance to birds or identification of individual birds are not required. | Indices only, and the relationship with true density of rock wrens would not be known, unless calibrated with known population size and the relationship doesn't change over time. Unadjusted for detectability, which will change with season, habitat and observer (though can be minimised through careful design and analysis). May not reflect subtle changes in populations. | Low. Set up cost is minimal. | [20,27] |

[a]See Methods text and references included within this table for definitions of techniques.

5MBC = 5-minute bird counts.

behaviour in relation to the nesting season and because it is difficult to compare with other methods (Table 1).

## Study areas

Rock wrens were monitored during the austral spring, summer and autumn seasons (October-May) in two alpine study areas: the Homer-Gertrude cirque in the head of the Hollyford Valley, Fiordland and Haast Range, South Westland in the Southern Alps of New Zealand [12]. The Homer-Gertrude cirque (44˚ 45' S, 168˚ 0' E), is a vertical sided U-shaped valley in the Darran Mountains at the head of the Hollyford Valley. Rock wren territories were located primarily in extensive boulder fields and talus slopes interspersed with subalpine scrub and patchy *Chionochloa* grasslands 700–1100 m a.s.l. The Haast Range study area, between the Waiatoto and Arawhata Rivers, was centred on Lake Greaney (44˚ 5' S, 168˚ 47'E). The study area was dominated by *Chionochloa* grasslands interspersed with scrub-covered cliff systems and the occasional boulder patch and talus slope, 1000–1400 m a.s.l.

## Count methods

We tested four counting methods in each of six breeding seasons between 2012 and 2018, and three sampling periods within each of the first four breeding seasons. Sampling periods were defined as: (1) nesting, when birds were nest building, incubating or feeding chicks on the nest (October-December), (2) fledging, once nestlings had left the nest (January-February) and (3) post-fledging (March onwards). Data from all four methods were collected during the fledging period annually during this period; however, data were not collected annually for all methods in the nesting and post-fledging periods due to logistical constraints and issues with detectability of rock wrens (see Results for details).

**Ethical statement.**   Rock wrens were banded by qualified banders under New Zealand's National Bird Banding Certification process. Because all other animal data collected in this study were strictly observational (i.e. no other animal handling occurred), an Animal Ethics permit was not required. Rock wrens are a protected species in New Zealand and the work occurred in two of the country's National Parks. Management of both the species and the Public Conservation Land in which they reside rests with our employer, Te Papa Atawhai—Department of Conservation. As such, a Wildlife Act permit was not required for the research.

**Territory mapping.**   Territory mapping was undertaken as a benchmark against which to compare population size with other count methods [8,20]. Throughout the study, rock wrens were uniquely colour banded. As many birds as possible were banded, with most banding conducted at the beginning of each breeding season, but with further birds marked opportunistically throughout (including fledglings as they became available). Sightings of known individuals, and their unbanded mates or family members, were collected throughout the nesting and fledging periods so that territories could be mapped. The locations of all birds were recorded with GPS (Garmin GPSMAP 64s) by field teams, generally of 2–4 people, whenever birds were encountered, and records entered into a sightings database. Locations of birds were also recorded on maps in the field for visualisation and cross-checking. Field teams searched the study areas for birds and nests on a weekly basis through the breeding season. They spent time following birds, identifying nesting sites and recording nesting phenology every few days, to get an idea of the core areas and limits of their ranges by recording multiple GPS locations. These maps distinguished between adult males and females, which have subtly different plumages [28] and fledglings.

**Mark-resight.**   Analysis of the number of marked animals seen within a population over multiple resighting surveys allows abundance to be estimated using the ratio of marked to unmarked individuals, if >40% of individuals in a population are marked [29–31]. Resightings of colour-banded and unmarked birds throughout the study areas were recorded during the three sampling periods (nesting, fledging and post-fledging) each breeding season. Survey routes

covered all rock wren territories mapped in the study areas to ensure equal probability of detection for all individuals and included the line transect routes used for other counts (see below). The observers walked the routes slowly four times during each sampling session in fine weather. If birds were detected, they were followed until it could be confirmed if they were banded or not.

**Simple counts on transects.** Simple counts were conducted along multiple 250 m line transects located randomly within suitable rock wren habitat in each study area. We defined suitable habitat as boulder fields, talus slopes, cliff systems, subalpine scrub based on mapping in ArcGIS Version 9 using SPOT satellite imagery. The number of transects sampled were representative of rock wren habitat in each study area: 14 transects at Homer-Gertrude cirque and 27 at Haast Range. Each transect was selected sequentially from a paired list of randomised start points and bearings, with a minimum of 250 m between transects. The observer walked slowly along the transect, recording all rock wrens seen or heard. Each transect took ca. 20 minutes to walk (mean = 22.5 minutes/transect). To maximise detectability of rock wrens, each transect was counted four times in fine weather (no rain, thick fog or strong winds), generally on consecutive days and each time by a different observer, where possible. Surveys took 3–8 days, depending on interruptions during adverse weather.

**Distance sampling.** Distance sampling, either from transects or points, is a widely used method for estimating abundance by modelling the detectability of an animal as its distance from the observer increases [3,9,29]. Rock wren counts from line transects described above were also used in calculating population estimates using distance sampling. In addition, perpendicular distance from the birds' location when first detected to the transect line was estimated visually or using Bushnell Yardage Pro 500 rangefinders to the nearest metre.

**Co-variates.** We recorded weather and environmental variables during each transect count. Variables included were: (1) temperature: cold = 0–5˚C, cool = >5–11˚C, mild = >11–-16˚C, warm = >16–22˚C, hot = >22˚C; (2) visibility at ground level in metres, averaged over the count; (3) overhead sunshine in minutes; (4) precipitation: 0 = none, 1 = misty, 2 = drizzle, 3 = light, 4 = moderate, 5 = heavy; and (5) wind: 0 = leaves still or moving without noise, 1 = leaves rustling, 2 = leaves and branchlets in constant motion, 3 = branches or trees swaying. These covariates were tested as predictors of rock wren detections on line transects (see below).

## Analysis

**Territory mapping.** Field maps from each field trip were checked against data entered in our resighting and nesting databases and combined to create single territory maps for the nesting and fledging periods each breeding season. Numbers of adults and fledglings and the territories they occupied were summed to estimate population size for each sampling period in each study area.

**Mark-resight.** The number of banded birds available for resighting was calculated from the territory maps and sightings database for each season, adjusted for additional birds banded between surveys and any deaths detected while monitoring nests. Data were analysed using the mark–resight modelling program NOREMARK for closed populations [30,31]. We used Bowden's estimator to compute mark–resight abundance estimates [30]. Bowden's estimator assumes that the probability of capture and resight is the same for all animals, but relaxes assumptions that the population is closed; allowing temporary movement out of the study area (in this case over inaccessible cliff edges), variation in resighting probabilities, sampling with replacement, and not requiring all animals to be correctly identified during each sampling session [9,32]. Although other models often appear to have greater precision, they can be overly precise and perform poorly when estimating confidence intervals compared with the Bowden's estimator [9,33].

**Simple counts on transects.** Despite constraining transect counts to relatively fine weather (no rain, thick fog or strong winds), we tested the effects of climatic variables on rock wren detections to evaluate the need for adjusting indices to account for any such effects. To do this we used a zero-inflated generalised mixed-effects model in RStudio (version 1.1.423) for a reduced dataset for which all covariate data were available (see Results). We included the following in the models: (1) rock wrens seen or heard on transects as the response variable, (2) temperature, sun (% direct sunshine during transect count), visibility, precipitation level and wind as fixed factors, and (3) transect nested within study area as the random effect. For comparison with other methods, rock wren detections on line transects were summarised as detections per transect per season for each year (Table 2).

**Distance sampling.** Data obtained by distance sampling were analysed using the program Distance 6.2 [34]. Observed differences in the general topography, vegetation composition and structure between the two study areas meant that detection probabilities for rock wrens at each study area were likely to differ. Data were therefore analysed independently for each study area [26]. As distances to birds were recorded to the nearest metre, distances were left ungrouped rather than being aggregated into distance classes.

To increase sample size and estimate precision, data from all surveys at each study area were pooled and global detection functions were calculated for each study area [26]. Using these global detection functions, data were post-stratified by survey and histograms of perpendicular distance measurements constructed. A selection of robust models and appropriate expansion functions recommended by [26] were then fitted. Model fit was assessed using

**Table 2. A comparison of rock wren, *Xenicus gilviventris*, estimates and indices of abundance from four monitoring methods at two study areas in the Southern Alps of New Zealand, 2012–2018.**

| Study area | Year | Sampling period | Territory map | Mark-resight (± 95% CI) | Distance sampling (± 95% CI) | Simple counts (mean ± SE) |
|---|---|---|---|---|---|---|
| Haast | 2012 | Nesting | 76 | 81 (64–102) | 102 (79–133) | 0.54 (± 0.09) |
| | 2013 | | 61 | 80 (60–107) | 13 (9–18) | 0.28 (± 0.06) |
| | 2014 | | 58 | 74 (55–100) | 9 (6–14) | 0.21 (± 0.05) |
| | 2015 | | 43 | 59 (36–100) | 35 (23–53) | 0.22 (± 0.07) |
| | 2013 | Fledging | 108 | 90 (71–113) | 113 (91–139) | 0.61 (± 0.08) |
| | 2014 | | 87 | 98 (68–142) | 14 (10–19) | 0.25 (± 0.06) |
| | 2015 | | 84 | 116 (71–191) | 8 (5–12) | 0.29 (± 0.06) |
| | 2016 | | 67 | 226 (108–474) | 52 (35–79) | 0.31 (± 0.06) |
| | 2017 | | 88 | 226 (123–418) | 39 (29–51) | 0.69 (± 0.12) |
| | 2018 | | 253 | 285 (244–333) | 44 (36–54) | 0.99 (± 0.12) |
| Homer/ | 2012 | Nesting | 56 | 66 (53–82) | 38 (16–69) | 0.56 (± 0.19) |
| Gertrude | 2013 | | 67 | 94 (64–138) | 42 (17–75) | 0.39 (± 0.11) |
| | 2014 | | 36 | 35 (26–47) | 68 (25–116) | 0.73 (± 0.15) |
| | 2015 | | 69 | 66 (52–83) | 82 (46–126) | 0.90 (± 0.14) |
| | 2013 | Fledging | 34 | 35 (30–41) | 40 (19–64) | 0.45 (± 0.09) |
| | 2014 | | 83 | 93 (62–142) | 61 (32–94) | 0.53 (± 0.14) |
| | 2015 | | 76 | 87 (64–119) | 55 (30–85) | 0.69 (± 0.14) |
| | 2016 | | 135 | 209 (136–332) | 212 (128–311) | 2.38 (± 0.29) |
| | 2017 | | 92 | 168 (101–282) | 28 (18–39) | 1.20 (± 0.18) |
| | 2018 | | 129 | 120 (98–147) | 24 (14–39) | 1.11 (± 0.23) |

Note that a major predation event at The Homer-Gertrude cirque in spring 2012 that resulted in failure of 100% of nests monitored and mortality of several adult females on nests [12] coincided with the monitoring comparison; as such these data should be treated with caution. The post-fledging sampling period was omitted due to low detection rates during this time (see Results).

Akaike's Information Criterion, Goodness of Fit and Q-Q plots and associated statistics, and the most parsimonious model was selected [6,26,35].

For the Homer-Gertrude study area, good model fit was achieved using half-normal and hazard rate models, and in the Haast Range, uniform and hazard rate models with varying numbers of adjustment terms in both areas (Table 3). The largest five percent of distance measurements were truncated to improve estimate precision [26].

**Comparison of methods.** We used linear models in RStudio to evaluate the relationship between estimates from territory maps and the other estimates/indices. We initially constructed global linear models to evaluate the influence of study area (Haast or Homer-Gertrude) and sampling period (nesting or fledging) in addition to each method (mark-resight, distance sampling, simple counts; separately) on estimates from territory maps. We then used linear models to explore correlations between territory maps and the other methods during the fledging period (for which most data exist; Table 2) for each study area separately.

**Estimating costs of methods.** By far the greatest cost involved in monitoring rock wrens was the wages paid to field staff. As such, we recorded the number of days and number of people required to undertake each technique as the basis for a cost comparison between methods. We assumed an hourly pay rate of NZD50 as the basis for calculations. We separated costs into: (1) costs associated with setting up the monitoring technique in year 1, (2) annual maintenance costs, if applicable to the method, and (3) annual monitoring costs.

## Results

We collected rock wren sightings data to create territory maps and compare these with population estimates from mark-resight and distance sampling data, and with indices from simple counts, across six years, 2012–2018 (see S1 Table for raw data). Robust data were collected in all six years during the fledging period (January–February) and in four of six years during the nesting period (October–December). However, detection rates in post-fledging period (March onwards) were too low to create meaningful estimates from territory maps (our 'gold

**Table 3. Models evaluated to produce population estimates from distance sampling.**

| Location | Model (key +adjustment)[1] | No. parameters | ΔAIC[2] | $\hat{D}$ensity (ha$^{-1}$) | 95% Confidence Interval | %CV[3] |
|---|---|---|---|---|---|---|
| Haast | Uniform + simple poly. (Poisson)* | 2 | 0 | 0.080 | 0.072–0.090 | 5.7 |
| | Uniform + cos. | 1 | 0.909 | 0.080 | 0.069–0.093 | 7.4 |
| | Hazard rate + simple poly. | 2 | 1.248 | 0.077 | 0.065–0.091 | 8.5 |
| | Hazard rate + cos. | 2 | 1.248 | 0.077 | 0.065–0.091 | 8.5 |
| | Half normal + hermite poly. | 1 | 4.091 | 0.082 | 0.070–0.096 | 7.9 |
| | Half normal +cos. | 1 | 4.091 | 0.082 | 0.070–0.096 | 7.9 |
| Homer-Gertrude | Half normal +cos. (Poisson)* | 3 | 0 | 0.304 | 0.256–0.360 | 8.7 |
| | Hazard rate + simple poly. | 2 | 1.787 | 0.291 | 0.238–0.354 | 10.1 |
| | Hazard rate + cos. | 2 | 1.787 | 0.291 | 0.238–0.354 | 10.1 |
| | Uniform + cos. | 1 | 3.726 | 0.251 | 0.225–0.281 | 5.7 |
| | Half normal + hermite poly. | 1 | 4.962 | 0.247 | 0.219–0.278 | 6.0 |
| | Uniform + simple poly. | 2 | 6.120 | 0.248 | 0.220–0.280 | 6.1 |

*Global model with lowest ΔAIC selected to compute post-stratified seasonal estimates in Table 2.

[1]Model consisting of a key function and an adjustment term.

[2]AIC values rescaled as simple differences between models.

[3]% Coefficient of Variation.

standard'), and so we excluded this period from analyses. Sample sizes presented below reflect only data collected in the nesting and fledging periods.

## Territory mapping

Four hundred and eighty birds were colour banded during the study (Homer-Gertrude = 197; Haast = 283). Although not all birds in the study areas were marked, the proportion of banded birds was sufficient to map territories and distinguish unbanded birds and breeding pairs, based on behaviours of marked and adjacent pairs, locations of nests, and behaviours at nests. Numbers of rock wrens counted each season varied between 34 and 135 adult birds at Homer-Gertrude and 43 and 253 at Haast (Table 2). The numbers of breeding rock wrens in the Haast study area declined over the first four breeding seasons (Fig 1A). However, following instigation of alpine predator trapping targeted at stoats (*Mustela erminea*) part way into the 2015/16 breeding season, productivity increased considerably, contributing to an increase in the study population over the following two seasons [12]. In contrast, the number of breeding birds at The Homer-Gertrude cirque increased steadily, though with some variability (Fig 1B), through the monitoring period during which time nests were protected using a similar predator trapping programme [12].

## Mark-resight

We undertook 10 mark-resight surveys during nesting and fledging in each study area across six breeding seasons, with equal survey effort in each study area. The proportion of each population banded (available for resighting) in each sampling period averaged 54.5 ± 17.3% SD (ra = 35.8–79.4%) at Homer-Gertrude and 50.9 ± 8.5% (ra = 41.1–65.8%) at Haast. Of these banded birds, on average, 50.1 ± 8.5% SD (Homer-Gertrude) and 49.0 ± 6.85% (Haast) were resighted on mark-resight surveys during nesting, with slightly lower proportions resighted during fledging (44.1 ± 14.8% and 30.6 ± 18.3% respectively).

Accuracy of recording band combinations was difficult to measure, but the rates of known partial or incorrect band combinations recorded give confidence that the majority of band combinations were recorded correctly. On average 3.6 ± 7.6% SD of sightings at Homer-Gertrude and 1.4 ± 3.4% of sighting at Haast were partial band combinations and incorrect band combinations were recorded on five (5.3%) of the mark-resight surveys (both study areas combined).

Population estimates derived from mark-resight data using Bowden's estimator ranged from 59 to 285 at the Haast study area and from 35 to 209 at the Homer-Gertrude cirque (Table 2). Precision of these estimates were highly variable in both study areas, with 95% confidence intervals being largest when population estimates were highest (Table 2).

## Distance sampling

Field observations showed that rock wren behaviours almost certainly violated assumptions of distance sampling (Table 1), particularly: (1) birds moving towards or away from the observer, and (2) all birds on the line are detected (i.e. birds obscured beneath large boulders on the line). The number of distance measurements to individual rock wrens for any given survey were highly variable, ranging from 14–120 birds at Homer-Gertrude and 18–110 birds for Haast Range. This corresponded to an encounter rate on transects of 1.0–9.2 birds detected/km surveyed at Homer-Gertrude and 0.2–2.4 birds/km at Haast. Pooling detections for each study area (assuming detectability at each study area remains the same over time), applying a global detection function, and post-stratifying by survey period provided a partial solution to

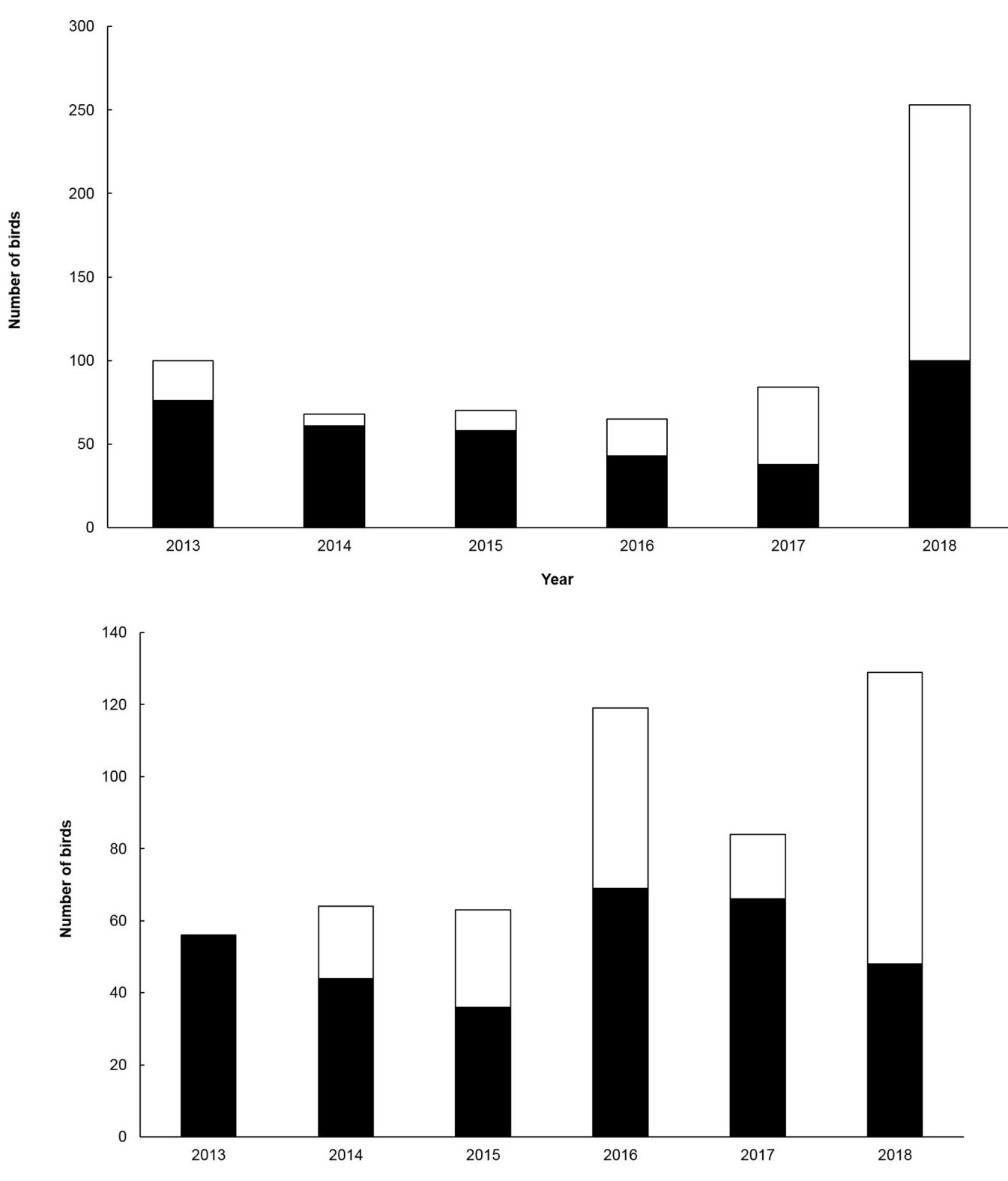

**Fig 1. Numbers of adult and fledgling rock wrens, *Xenicus gilviventris*, detected through territory mapping during the fledging (January-February) period 2012–2018 at: (a) the Haast Range, South Westland, and (b) the Homer-Gertrude cirque, Fiordland.**

the lack of data for some surveys. However, extremely low sample sizes in some surveys compromised the precision of abundance estimates [26] (Table 2).

Abundance estimates and corresponding 95% confidence intervals (only bootstrapped for the Homer-Gertrude study area because estimates for Haast failed to converge) ranged from 8 to 113 for Haast and from 24 to 212 in Homer-Gertrude (Table 2 and Fig 2). Confidence Intervals for some survey periods were extremely wide for both study areas, particularly for surveys where modelled estimates of abundance were large (Fig 2).

### Simple counts on transects

Between 2012 and 2018 we recorded 968 rock wrens (i.e. birds seen or heard) during 1647 transect counts conducted in the nesting and fledging periods. Indices of abundance (i.e. number of birds detected per transect during a sampling period) ranged from 0.21 to 0.99 birds per transect at Haast and 0.39 to 2.38 birds per transect at Homer-Gertrude (Table 2).

The full range of weather variables was measured on 922 transect counts; these counts were used to evaluate relationships between rock wren counts and weather variables (air temperature, sunshine, visibility, precipitation and wind). Within the sampling constraints imposed (counts were conducted during fine weather; that is, no rain, thick fog or strong wind), we detected no significant correlations between rock wren counts and weather recorded ($P > 0.05$ in all cases). As such, we did not need to adjust the indices of abundance obtained from line transect sampling for weather variables prior to comparison with estimates derived from territory mapping.

### Comparison of techniques

Estimates of rock wren populations based on territory mapping during the fledging period (January-February) were positively correlated with indices of abundance from simple counts on line transects at both study areas during the same period (Haast: $t_1 = 3.041$, $P = 0.038$; Homer-Gertrude: $t_1 = 2.555$, $P = 0.063$; Table 4 and Fig 3). These territory map estimates were also positively correlated with estimates derived from mark-resight surveys in the Homer-Gertrude cirque during the fledging period ($t_1 = 2.922$, $P = 0.043$), but not at the Haast study area ($t_1 = 1.351$, $P = 0.248$). However, rock wren population estimates from distance sampling were not correlated with estimates from territory maps in either study area ($P > 0.1$ for both study areas; Table 4 and Fig 3).

**Estimating costs of methods.** Set-up costs for the methods requiring banding rock wrens (territory mapping and mark-resight) were 25 times higher than methods for which birds were not banded (distance sampling and simple counts; Table 5). Annual costs (maintenance and monitoring costs combined) were greatest for territory mapping at NZD 22,000 p.a., intermediate for mark-resight at NZD 13,200 p.a. and lowest for distance sampling and simple counts, both of which costs NZD 3,200 p.a. (Table 5). As such, annual costs of territory mapping were 6.9 times higher than annual costs both distance sampling and simple counts.

### Discussion

The strong, positive correlations between estimates from territory maps (the 'gold standard' technique) and indices of relative abundance derived from simple counts on line transects suggest that these counts can be used as a low-cost, reliable, technique to monitor trends in rock wren populations over time. Banding birds (required for territory mapping and mark-resight) is time-intensive and dependent on the availability of skilled personnel to coincide with fair weather conditions, which can be problematic in the alpine zone of an oceanic island like New Zealand. We estimate that annual costs of territory mapping are 6.9 times higher, and annual

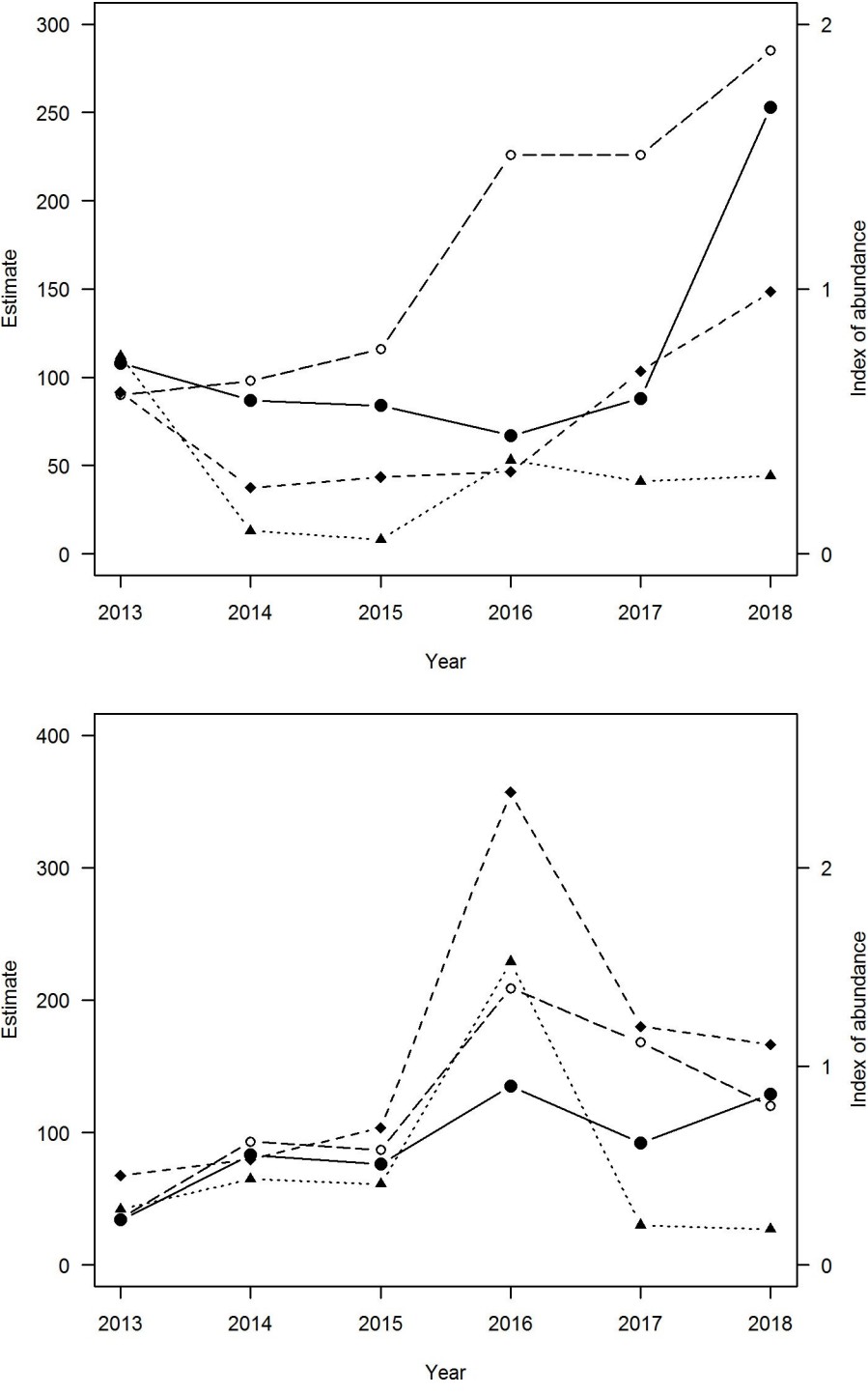

**Fig 2. Rock wren, *Xenicus gilviventris*, estimates and indices of abundance from four methods over six summer fledging periods at two alpine study areas ((a) Haast and (b) Homer-Gertrude) in the Southern Alps of New Zealand.** Symbols are as follows: (1) territory map = black circles; (2) mark resight = open circles; (3) distance sampling = black triangles; (4) simple counts = black diamonds. The left axis shows the territory map figures and estimates from mark resight and distance sampling; the right axis shows the index of abundance from line transects. See Table 2 for variance estimates.

**Table 4. Correlation coefficients (r² values from linear models) between abundance derived from territory mapping of rock wrens, *Xenicus gilviventris*, and population estimates from both mark-resight and distance sampling and indices of relative abundance from simple counts on line transects.**

| | Haast | | Homer-Gertrude | |
| --- | --- | --- | --- | --- |
| | Overall | Fledging period | Overall | Fledging period |
| Territory ~ mark-resight | 0.47 | 0.31 | 0.76 | 0.68 |
| Territory ~ distance sampling | 0.02 | 0.01 | 0.20 | 0.25 |
| Territory ~ simple counts | 0.72 | 0.70 | 0.58 | 0.49 |

Data from the nesting period are not presented separately due to insufficient data points (n = 4 per study area), but are included within the overall metrics.

costs of mark-resight are 4.1 times higher than those of simple counts and distance sampling. Because simple counts don't require banding of rock wrens, they provide a much more cost-effective monitoring tool that we have now validated against a benchmark. We do, however, recommend that results obtained through this method be interpreted with appropriate caution given that index methods lack incorporation of detection probabilities, which may vary across habitats, densities and time [36,37].

Although territory mapping is often considered the 'gold standard' for monitoring bird species [20], this technique still does not represent a true census. Because birds are highly cryptic, and frequently occupy inaccessible cliff habitats, it is difficult to accurately measure immigration, emigration and mortality during a season. Crypsis in rock wrens increases as the season wears on, with the proportion of birds detected in surveys in the post-fledging period being markedly lower than in both the nesting and fledging periods due to fledglings becoming independent and dispersing throughout the landscape [38], as has been previously reported for house wrens, *Troglodytes aedon*, in Ohio, USA [39]. Nevertheless, we felt population estimates derived from territory mapping in the nesting season were accurate, and changes in population sizes were what was expected given our monitoring of high predation rates, particularly by invasive stoats, prior to introduction of population scale trapping and documented recovery afterwards at both study areas [12]. We were unable to obtain sufficient data for all methods trialled to undertake a statistical comparison of methods during the post-fledging period. However, based on the limited data we were able to collect during this period, we suggest that results from all methods are more variable and less reliable in the post-fledging period than earlier in the breeding season. The increased crypsis later in the season suggests that monitoring of trends in rock wrens should be timed consistently each year and occur prior to dispersal of fledglings.

Accuracy and precision of population estimates derived using Bowden's mark-resight estimator were highly variable. Incomplete identification of marked individuals is potentially a major source of bias in mark-resight abundance estimators [40] applicable to rock wrens because full colour band combinations are not always seen when rock wrens are only glimpsed briefly. Population estimates generated from mark-resight data from visual surveys were only strongly correlated with estimates from territory mapping at one of our two study areas. Rock wrens were highly cryptic, with a low proportion detected on any one survey. They often fed underground in the extensive boulder fields and dense subalpine scrub that characterises their habitat. Further, mark-resight surveys in the alpine zone were very labour intensive. In addition to the considerable effort involved in banding a meaningful proportion of the population, the thorough surveys required to sample all territories in the population were very extensive. For example, the route surveyed in the Haast Range was c. 35 km long to sample c. 30 rock wren territories. Thus, increasing the effort to complete more than four surveys to further increase the proportion of marked birds detected and the accuracy of the technique would use considerable additional resources with no guarantee of achieving this objective.

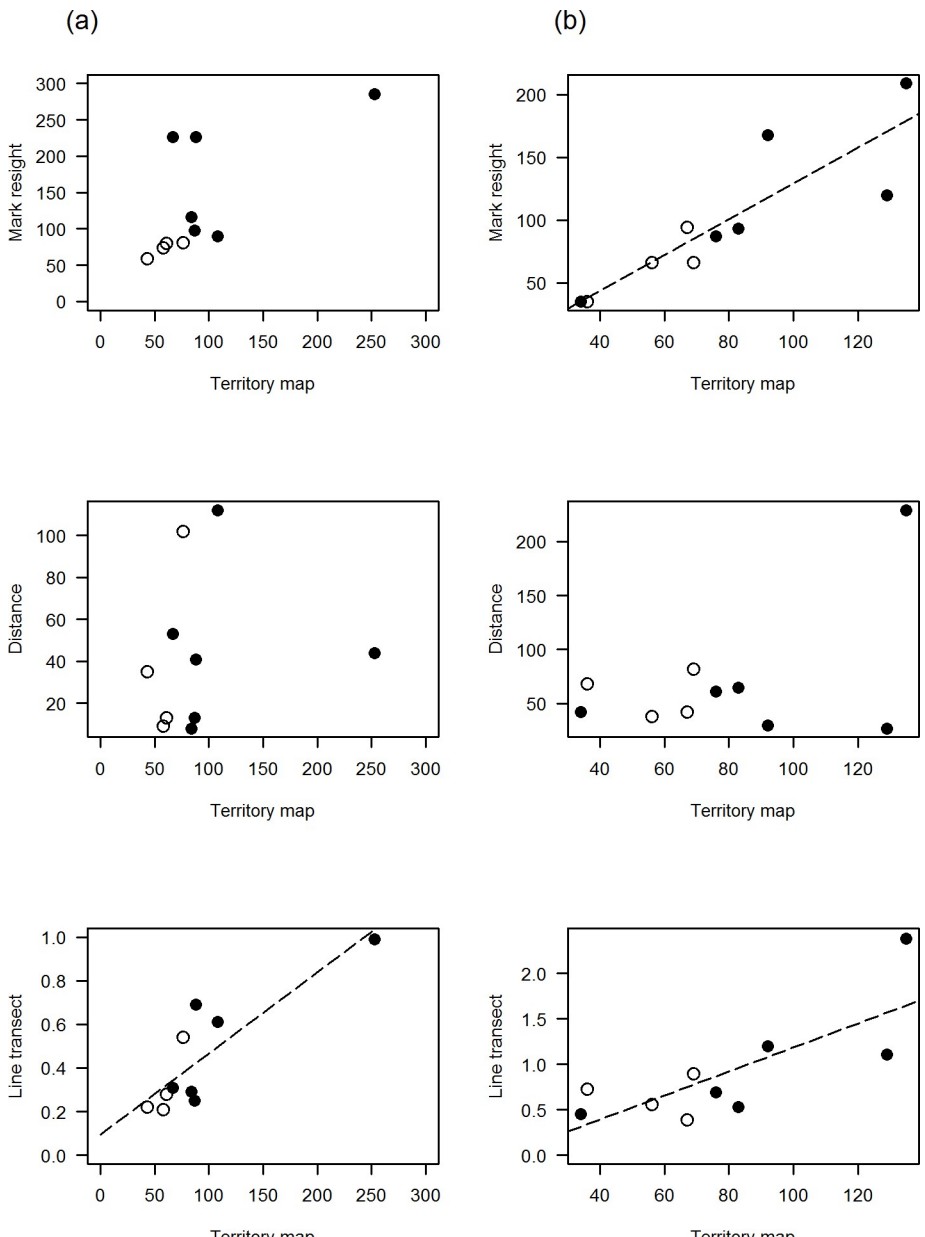

**Fig 3. Comparison of three methods (mark-resight, distance sampling and line transects) against territory mapping for rock wrens, *Xenicus gilviventris*, at the (a) Haast and (b) Homer-Gertrude alpine study areas in the South Island of New Zealand.** Filled circles = summer; open circles = spring; dashed regression line added for significant linear relationships between methods (see Results).

Distance sampling has recently been used successfully to estimate densities of several forest birds and to evaluate their long-term responses to conservation management [3,41]. However, population estimates for rock wrens derived from distance sampling were not correlated with population estimates produced from territory maps. At least two of the three key assumptions of distance sampling (Table 1) are regularly violated in rock wren monitoring. Firstly, the assumption of 100% detectability on the zero line (i.e. the transect) was not achieved where large boulders with sub-terranean space were present on the transect line and rock wrens

**Table 5. Cost comparison for four monitoring methods for rock wrens, *Xenicus gilviventris*.**

| Technique | Set-up | | | Annual maintenance | | | Annual monitoring | |
|---|---|---|---|---|---|---|---|---|
| | Tasks | Time | Cost | Tasks | Time | Cost | Time | Cost |
| Territory mapping | Scope study area Catch initial sample of birds | 10 days x 5 people | $20,000 | Keeping a sample of birds tagged | 5 days x 5 people | $10,000 | 15 days x 2 people | $12,000 |
| Mark-resight | Scope study area Catch initial sample of birds | 10 days x 5 people | $20,000 | Keeping a sample of birds tagged | 5 days x 5 people | $10,000 | 4 days x 2 people | $3,200 |
| Distance sampling | Plan & mark transects | 2 person days | $800 | None | | $0 | 4 days x 2 people | $3,200 |
| Simple counts | Plan & mark transects | 2 person days | $800 | None | | $0 | 4 days x 2 people | $3,200 |

All costs are presented in New Zealand Dollars (NZD).

frequently used that space to forage or take refuge, without making many calls. This regularly occurred when attempts were made to catch rock wrens for banding or when rock wrens were nesting beneath boulders. Secondly, the assumption that birds do not approach or avoid observers was violated in open habitat types where the birds frequently flew away from observers on approach. The poor performance of distance sampling as a technique for producing population estimates in rock wrens is similar to that seen in bellbirds, *Anthornis melanura*, which were also variable in conspicuousness and moved away from the line transect when approached by an observer [41]. However, it is somewhat surprising that distance sampling performed so poorly in comparison to indices of relative abundance which were derived from the same line transects. One potential explanation is that the estimated distances were so inaccurate that including them in the population estimation process introduced more error than it removed. Initial attempts to use range finders to measure distance were often thwarted by misty conditions in the alpine zone and we resorted to visual estimation of distance in most cases. Further, it is possible that systematic bias was induced by estimated distances being correlated with habitat.

More promisingly, indices of abundance generated from the simple counts on line transects were strongly correlated with territory map estimates at both of our alpine study sites over a six-year period. Repeated sampling on line transects has shown similar promise in open fen mire habitat for aquatic warblers, *Acrocephalus paludicola*, in Central Europe [4] and in forest for endemic passerines, *Mohoua ochrecephala*, in New Zealand [42]. In the latter study, monitoring of 14 *Mohoua ochrecephala* populations on line transects at 12 sites over up to 11 years revealed one population extinction and a further five populations in decline [42]. This led to an understanding that a species previously thought to be secure was endangered and in need of immediate conservation intervention [42]. When applying our findings to recommending a new standard monitoring technique for rock wrens we acknowledge that the relationship between the index and real density may not remain the same over time. Lower correlation coefficients at the Homer-Gertrude study area can be partially attributed to the index overestimating the population in the fledging period of 2016 when the population was at its highest during the sampling period (2012–2018). This may hint at a non-linear relationship between detectability and density whereby birds are disproportionately active and vocal as density increases. A non-linear relationship between detectability and density was also observed for South Island robins, *Petroica australis*; however, in the case of robins, detectability increased at lower population density due to an increase in calling by males when females are scarce in the population [9]. Therefore, while we recommend that indices of abundance on line transects be adopted as a standard low-cost technique for monitoring trends in rock wren populations, we

suggest that care be taken in interpreting results derived from indices, for which detection probabilities are not accounted [36,37]. Further targeted testing of the line transect method at even higher densities should be undertaken if populations increase beyond levels observed during this study. Promisingly, this seems likely for populations where effective alpine predator control in the form of population-scale trapping [12] or landscape-scale toxin application [43] leads to further increases in rock wrens.

## Conclusion

Monitoring population trends in montane and alpine birds is becoming increasingly important to determine potential impacts of climate change and increased anthropogenic disturbance [37,44–46]. In general, calibrating bird monitoring methods has been undertaken infrequently, limiting comparability among monitoring programmes [47,48]. Our comparison of monitoring methods for alpine passerines in New Zealand, and finding that a low-cost index technique is strongly correlated with estimates from territory mapping which is regarded as a 'gold standard' in bird monitoring, has applicability to other open habitat bird species as well as highlighting the importance of validating potential monitoring techniques, prior to using them as the basis of monitoring protocols.

## Supporting information

**S1 Table. Raw data collected on rock wrens, *Xenicus gilviventris*, using four techniques (territory mapping, mark-resight, distance sampling and simple counts) at two sites in the Southern Alps of New Zealand, 2012–2018.**
(XLSX)

## Acknowledgments

Thanks to Sue Heath, Bruce Robertson, Megan Willans and James Reardon for input into study design and Richard Earl for assistance with mapping rock wren habitat in the study areas. Thanks also to Kathrin Affeld, Will Batson, Becky Bell, Crystal Brindle, Iris Broekema, Bevan Cameron, Jo Carpenter, Jono Dobbs, Phil Evans, Clare Kilner, Ian Clark, Eric Edwards, Sarah Forder, Ruth Garland, Flo Gaud, Lynette Hartley, Adam Ingram, Athene Irvine, Rebecca Jackson, Franziska Landesberger, Rose Lanman, Jamie McAulay, Bruce McKinlay, Fraser Maddigan, James Maunder, Grant Maslowski, Kathy Morrison, Dan Palmer, Moira Pryde, Emma Richardson, Bruce Robertson, Lucy Rossiter, Sam Rowland, Anne Schlesselmann, Sanjay Thakur, Jo Tilson, Kerry Uren, Antje Wahlberg, Jim Watts, David Webb, Jemma van Beek, Maddie van de Wetering, Jason van de Wetering, Megan Willans, Kailash Willis and Rebecca Wilson for assistance conducting counts in the field and banding rock wrens, Kathrin Affeld for data management, Karina Sidaway for technical editing and the staff of the Haast and Te Anau Department of Conservation offices for considerable assistance with logistics.

## Author Contributions

**Conceptualization:** Joanne M. Monks, Colin F. J. O'Donnell, Terry C. Greene, Kerry A. Weston.

**Data curation:** Joanne M. Monks, Colin F. J. O'Donnell, Kerry A. Weston.

**Formal analysis:** Joanne M. Monks, Colin F. J. O'Donnell, Terry C. Greene, Kerry A. Weston.

**Funding acquisition:** Joanne M. Monks, Colin F. J. O'Donnell.

**Investigation:** Joanne M. Monks, Colin F. J. O'Donnell, Kerry A. Weston.

**Methodology:** Joanne M. Monks, Colin F. J. O'Donnell, Terry C. Greene.

**Project administration:** Kerry A. Weston.

**Writing – original draft:** Joanne M. Monks, Colin F. J. O'Donnell, Kerry A. Weston.

**Writing – review & editing:** Joanne M. Monks, Terry C. Greene.

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
