## [Decision Letter · Decision Letter 0]

4 Jan 2021

PONE-D-20-38105

Evaluation of counting methods for monitoring populations of a cryptic alpine passerine, the rock wren (*Xenicus gilviventris*)

PLOS ONE

Dear Dr. Monks,

Thank you for submitting your manuscript to PLOS ONE. After careful consideration, we feel that it has merit but does not fully meet PLOS ONE’s publication criteria as it currently stands. Therefore, we invite you to submit a revised version of the manuscript that addresses the points raised during the review process.

We look forward to receiving your revised manuscript.

Kind regards,

Bi-Song Yue, Ph.D

Academic Editor

PLOS ONE

2. We note that Figure 1 in your submission contain map images which may be copyrighted. All PLOS content is published under the Creative Commons Attribution License (CC BY 4.0), which means that the manuscript, images, and Supporting Information files will be freely available online, and any third party is permitted to access, download, copy, distribute, and use these materials in any way, even commercially, with proper attribution. For these reasons, we cannot publish previously copyrighted maps or satellite images created using proprietary data, such as Google software (Google Maps, Street View, and Earth). For more information, see our copyright guidelines: http://journals.plos.org/plosone/s/licenses-and-copyright.

2.1.    You may seek permission from the original copyright holder of Figure 1 to publish the content specifically under the CC BY 4.0 license. 

2.2.    If you are unable to obtain permission from the original copyright holder to publish these figures under the CC BY 4.0 license or if the copyright holder’s requirements are incompatible with the CC BY 4.0 license, please either i) remove the figure or ii) supply a replacement figure that complies with the CC BY 4.0 license. Please check copyright information on all replacement figures and update the figure caption with source information. If applicable, please specify in the figure caption text when a figure is similar but not identical to the original image and is therefore for illustrative purposes only.

Reviewers' comments:

Reviewer's Responses to Questions

**Comments to the Author**

1. Is the manuscript technically sound, and do the data support the conclusions?

Reviewer #1: Yes

Reviewer #2: Partly

2. Has the statistical analysis been performed appropriately and rigorously? 

Reviewer #1: Yes

Reviewer #2: Yes

3. Have the authors made all data underlying the findings in their manuscript fully available?

Reviewer #1: Yes

Reviewer #2: Yes

4. Is the manuscript presented in an intelligible fashion and written in standard English?

Reviewer #1: Yes

Reviewer #2: Yes

5. Review Comments to the Author

Reviewer #1: This is a well-done study that evaluates different monitoring methods and their appropriateness for a particular species of wren in New Zealand. The authors have done an excellent job conveying what they did and its importance for species conservation. I have a few remaining questions and suggestions.

- I (not an ornithologist) struggled to understand why the numbers in the simple count method were given as abundance indices, and not converted to a number that would be directly comparable to the territory mapping. Would doing so have altered the statistical results in a meaningful way?

- Line 84: I’d like a lot more detail about this literature review, particularly since it is called a lit review. Are these methods that are usually used in alpine bird species? Was anything in particular discarded, and if so, why?

- In table 1, please add a column with a short description of each method. This information is in the body of the text but not the first time these terms are used.

- Why have the methods been ordered this way in the table? As a reader, it seems to make more logical sense that territory mapping would be described first or last (as the standard to which all other methods are compared)

The line numbers disappeared after Table 1.

- In the Distance Sampling methods paragraph (page 10), please explain what conventional methods are in a short sentence or two.

- Page 11 – NOREMARK is a different spelling than in Table 1. I think this spelling is correct and the table is wrong, but please check.

- In the results, page 14, you mention that territory mapping is only reliable through the fledgling period. There is a bit of discussion of this issue on page 19, but I’d like more – is this expected to be true of all of these methods? Might one method be better for surveys in the post-fledgling period?

- Fig 4 – I don’t’ see an explanation of panel a vs b (they are clearly the two study sites but it’s not clear which is which).

Reviewer #2: Please include in the title the Order and the Family of the bird species.

This study departs from a relevant perspective and searching for cost-effective and cost-less methods is valid if your output is equally valid, replicable, and renders strong data to be used in long-term monitoring and conservation. Comparison between different methods is also relevant, and this study is particularly important in bringing some light on this neglected subject. One of the authors' concerns is about the cost of each method x quality of the results, and this statement appears along with the manuscript. However, this topic is not explored with the details requirable in the discussion, and according to the authors, in a situation where the results must be "achievable within budget constraints" I hope to see this topic explored with the necessary details. As the authors have the raw data & costs to obtain these data in each methodology I strongly suggest that they can cover this topic more accurately. I agree that banding birds is the most expensive method and potentially harmful specially for delicate and threatened species. Although I agree that pointing counts is an effective tool, I believe the authors must provide stronger evidence in the discussion (maybe the last paragraph should be subjected to a more lengthy explanation).

6. PLOS authors have the option to publish the peer review history of their article (what does this mean?). If published, this will include your full peer review and any attached files.

Reviewer #1: No

Reviewer #2: No

---

## [Author Response · Author response to Decision Letter 0]

14 Feb 2021

Bi-Song Yue

Academic Editor

PLOS ONE

Dear Dr Bi-Song Yue

Thank you very much for the constructive feedback on our manuscript. We have thoroughly addressed all points raised during review. Our detailed responses to reviewers’ comments are in italics below.

We hope that our revised manuscript is now suitable for publication in PLOS One and look forward to hearing from you again.

With best wishes,

Jo Monks (on behalf of all authors)

> Done. 

2. We note that Figure 1 in your submission contain map images which may be copyrighted. We require you to either (1) present written permission from the copyright holder to publish these figures specifically under the CC BY 4.0 license, or (2) remove the figures from your submission.

> On reflection, Figure 1 (a map showing location of the two study sites) is not essential to this manuscript, so we have decided to remove it rather than seeking copyright permission. Further, we now refer to another paper which includes a map showing site locations. 

> Done.

Reviewer #1: 

This is a well-done study that evaluates different monitoring methods and their appropriateness for a particular species of wren in New Zealand. The authors have done an excellent job conveying what they did and its importance for species conservation. I have a few remaining questions and suggestions.

- I (not an ornithologist) struggled to understand why the numbers in the simple count method were given as abundance indices, and not converted to a number that would be directly comparable to the territory mapping. Would doing so have altered the statistical results in a meaningful way?

> Simple counts are presented as abundance indices because the number of transects surveyed at each site differed according to availability of suitable habitat for rock wrens and, due to weather and logistical constraints, it was not always possible to survey all transects an equal number of times per sampling period. Furthermore, attempting to present simple counts as a number to be directly compared with estimates from territory mapping would be misleading, because they do not sample the full area and because they do not account for detectability.

- Line 84: I’d like a lot more detail about this literature review, particularly since it is called a lit review. Are these methods that are usually used in alpine bird species? Was anything in particular discarded, and if so, why?

> Reviewer 1 makes a good point that we didn’t conduct a formal literature review in creating this table, rather researched the applicability of established bird counting techniques to rock wrens and this is what we present in Table 1. We have removed reference to the term ‘literature review’.

> Based on our understanding from the literature, there is no ‘usual’ method for alpine bird species. Some of these techniques are applied elsewhere in the world in alpine, or more usually open, habitats, but we didn’t find strong evidence of consistency. This was part of our motivation for the present study, which we hope will contribute to the literature on methods for alpine birds.

> Site occupancy was discarded because detectability can vary with rock wren behaviour in relation to the nesting season and because it is difficult to compare with other methods. See lines 93 to 95 of the Methods for this explanation.

- In table 1, please add a column with a short description of each method. This information is in the body of the text but not the first time these terms are used.

> We attempted to do this, but it is impossible to describe these techniques briefly in a way amenable to inclusion in a table; the detail is important! We have instead referred the reader to the Methods text and references included within the table in the footnotes of the table for these descriptions.

- Why have the methods been ordered this way in the table? As a reader, it seems to make more logical sense that territory mapping would be described first or last (as the standard to which all other methods are compared)

> This is a good point. We have reordered the table to begin with territory mapping (as the gold standard), followed by the other techniques in decreasing order of cost.

The line numbers disappeared after Table 1.

> Good point! Now fixed.

- In the Distance Sampling methods paragraph (page 10), please explain what conventional methods are in a short sentence or two.

> Done. We’ve added a description of distance sampling and rewritten the paragraph for clarity.

- Page 11 – NOREMARK is a different spelling than in Table 1. I think this spelling is correct and the table is wrong, but please check.

> Thanks – we’ve corrected the spelling error in Table 1.

- In the results, page 14, you mention that territory mapping is only reliable through the fledgling period. There is a bit of discussion of this issue on page 19, but I’d like more – is this expected to be true of all of these methods? Might one method be better for surveys in the post-fledgling period?

> We have added the following two sentences to the paragraph Reviewer 1 refers to in order to address this point: 

“We were unable to obtain sufficient data for all methods trialled to undertake a statistical comparison of methods during the post-fledging period. However, based on the limited data we were able to collect during this period, we suggest that results from all methods are more variable and less reliable in the post-fledging period than earlier in the breeding season.”

- Fig 4 – I don’t’ see an explanation of panel a vs b (they are clearly the two study sites but it’s not clear which is which).

> Good point! We’ve added this information into the figure title.

Reviewer #2: 

Please include in the title the Order and the Family of the bird species.

> Done.

This study departs from a relevant perspective and searching for cost-effective and cost-less methods is valid if your output is equally valid, replicable, and renders strong data to be used in long-term monitoring and conservation. Comparison between different methods is also relevant, and this study is particularly important in bringing some light on this neglected subject. One of the authors' concerns is about the cost of each method x quality of the results, and this statement appears along with the manuscript. However, this topic is not explored with the details requirable in the discussion, and according to the authors, in a situation where the results must be "achievable within budget constraints" I hope to see this topic explored with the necessary details. As the authors have the raw data & costs to obtain these data in each methodology I strongly suggest that they can cover this topic more accurately. I agree that banding birds is the most expensive method and potentially harmful specially for delicate and threatened species.

> This is a very good point. In response we collated information on set-up, annual maintenance and annual monitoring costs for all four field methods. This information is now presented in a new table (Table 5), described in the methods and results section, and referred to in the discussion.

Although I agree that pointing counts is an effective tool, I believe the authors must provide stronger evidence in the discussion (maybe the last paragraph should be subjected to a more lengthy explanation).

> We have expanded this paragraph of the discussion to strengthen the evidence for utility of repeated counts on line transects as a suitable technique for trend monitoring in rock wrens and potentially other passerines in open habitat. In doing so, we made it explicit that in our study indices of abundance from line transects were strongly correlated with estimates from territory mapping at both study sites over a 6-year period. We also added a case study in which the technique was successfully applied to a forest-dwelling passerine and resulted in strong management recommendations.

---

## [Editor Report · Decision Letter 1]

16 Feb 2021

Evaluation of counting methods for monitoring populations of a cryptic alpine passerine, the rock wren (Passeriformes, Acanthisittidae, Xenicus gilviventris)

PONE-D-20-38105R1

Dear Dr. Monks,

We’re pleased to inform you that your manuscript has been judged scientifically suitable for publication and will be formally accepted for publication once it meets all outstanding technical requirements.

Kind regards,

Bi-Song Yue, Ph.D

Academic Editor

PLOS ONE

---

## [Editor Report · Acceptance letter]

23 Feb 2021

PONE-D-20-38105R1 

Evaluation of counting methods for monitoring populations of a cryptic alpine passerine, the rock wren (Passeriformes, Acanthisittidae, *Xenicus gilviventris*) 

Dear Dr. Monks:

I'm pleased to inform you that your manuscript has been deemed suitable for publication in PLOS ONE. Congratulations! Your manuscript is now with our production department. 

Kind regards, 

on behalf of

Dr. Bi-Song Yue 

Academic Editor

PLOS ONE